# Prenatal Diagnosis and Pregnancy Termination in Jewish and Muslim Women with a Deaf Child in Israel

**DOI:** 10.3390/children10091438

**Published:** 2023-08-23

**Authors:** Aliza Amiel, Wasef Na’amnih, Mahdi Tarabeih

**Affiliations:** 1School of Nursing Science, The Academic College of Tel Aviv-Yaffo, Tel Aviv 64044, Israel; wasef25@yahoo.com (W.N.); mahdita@mta.ac.il (M.T.); 2Department of Epidemiology and Preventive Medicine, School of Public Health, Sackler Faculty of Medicine, Tel Aviv University, Tel Aviv 69978, Israel

**Keywords:** hearing loss, Jewish and Muslim women, pregnancy termination, prenatal tests

## Abstract

Deafness is the most common sensory disability in humans, influencing all aspects of life, However, early diagnosis of hearing impairment and initiating the rehabilitation process are of great importance to enable the development of language and communication as soon as possible. We examined the differences in attitudes towards performing prenatal invasive tests and pregnancy terminations in Jewish and Muslim women in Israel due to deafness. Overall, 953 Israeli women, aged 18–46 years with a mean age of 32.0 (SD = 7.12), were enrolled. Of those, 68.7% were city dwellers and 31.3% were village dwellers, and 60.2% were Muslim women and 39.8% were Jewish women. All participants had a child with a hearing impairment or deafness. The group with no genetic hearing loss performed more prenatal invasive tests and pregnancy terminations than those with genetic hearing loss in both ethnic groups. Jewish women performed more invasive prenatal tests and, consequently, a pregnancy termination. Secular Jewish women more frequently underwent pregnancy terminations due to fetal deafness. Further genetic counseling and information concerning IVF and PGD procedures should be provided to the Muslim population.

## 1. Introduction

Deafness is a multifactorial varied trait [1,2], and the most common sensory disability in humans, influencing all aspects of life. Approximately 50% of congenital deafness is hereditary; the other 50% is still indeterminate. One in 1000 babies will be born completely deaf; one in 200 will be born with some type of hearing impairment [3]. At present, more than 150 genes have been known to cause hearing loss. Specific genes have been found in diverse populations, i.e., >20 genes cause deafness within the Jewish–Israeli hearing-impaired population. The most common gene found worldwide is the *GJB2* gene, which encodes the connexin 26 protein. The *GJB2* gene and the *TMC1* have both been found in Moroccan Jews [4]. 

*GJB2* variants are the most prevalent cause of hereditary hearing loss worldwide and are responsible for approximately 30% of deafness in Jewish families [3,5,6]. In Israel, mutations include some responsible for hearing loss, such as *GJB2 c.167delT* and *TMC1 p.Ser647Phe*, while other deafness-causing mutations, such as *GJB2 c.35delG*, are common in all Jewish ethnicities and elsewhere [5,7]. Deep sequencing was introduced less than a decade ago, and, subsequently, the number of deafness-related genes identified in all ethnic populations has increased by threefold. By identifying the pathogenic variant, the clinician can study the molecular pathogenesis, calculate the possibility of concomitant morbidity, offer a prenatal diagnosis, and make suggestions on the next step, regarding whether to continue or terminate the pregnancy. Presently, cochlear implants increase rehabilitation success. Understanding molecular pathogenesis will hopefully lead to personalized medical treatment [4]. In an extensive review of Muslim participants residing in 22 Muslim countries, 104 variants in 44 genes were found in 17 individuals. Of these variants, 72 (of the 41 genes) were unique to Muslim patients who manifested a distinctive clinical phenotype [8]. In a study performed on a Muslim population residing in northern Israel, the prevalence of genetic hearing loss (HL) was 47%, almost identical to that of the Jewish population despite Muslim consanguineous marriages [9]. It is recognized that pregnancy termination is forbidden in the Islamic religion beginning 120 days after gestation. Most Muslim and Druze academics agree that abortion is prohibited after the ensoulment of 120 days. Nonetheless, a dispute exists between some researchers as to when an abortion should be allowed (before the 120-day limit), for example, if the mother will die or will be severely injured at birth, or if the fetus exhibits serious hearing loss and no clinical treatment is accessible, or the fetus will be born with an incurable disease, making the future life of the mother and the child unbearable [10,11,12,13]. According to Islamic laws, HL is not considered a serious disability; hence, terminating a pregnancy is not permitted [10]. 

The importance of early identification of hearing impairment is well established. Cumulative evidence shows that undiagnosed or untreated permanent hearing impairment (PHI) during early childhood may result in speech–language delays, poor academic achievements, and social and emotional difficulties [14]. Such delays in the different domains have also been documented for those whose PHI was mild to moderate or only in one ear [15,16,17]. Currently, there is overwhelming evidence that early diagnosis and habilitation before the age of six months improves speech and language development and cognitive outcomes, reducing the need for special education and improving quality of life [18,19,20]. In order to identify hearing-impaired infants as early as possible and offer them the appropriate intervention, the National Institute of Health in 1993 recommended the implementation of universal neonatal hearing screening programs (NHSP) up to the age of three months in order to initiate hearing habilitation no later than the age of six months [21]. The Israeli national hearing screening program at that time was conducted at Mother and Child Health Clinics using the distraction test at ages 7–9 months. Since 1997, a number of medical centers in Israel began offering NHSP. The Ministry of Health Directive 33/2009 established the guidelines for the NHSP for all infants to be implemented from 1 January 2010 [20,21,22]. Following the recommendations stated in these guidelines, the current program consists of the Optoacoustic Emissions (OAE) test for all newborns, and the Automated Auditory Brainstem Response test was established for those infants who failed OAE testing and for infants at risk for auditory neuropathy spectrum disorders [23]. The national program objectives are to complete screening by the age of one month, conduct diagnostic audio-logical testing no later than the age of three months for those infants who failed the screening, and initiate habilitation for those diagnosed with hearing loss by the age of six months. Early screening covers all newborns in the country and continues for those who failed the secondary screens and those identified as high-risk groups [23]. Definitive diagnostic testing during pregnancy is an invasive procedure. Chorionic villus sampling (CVS) is performed between 10 and 13 weeks of development and amniocentesis (AC) between 16 and 20 weeks. Both procedures can cause a miscarriage with a risk of 0.5–1%, even though recent meta-analyses studies have proposed that the actual procedure-related risk may be much lower. Karyotyping or microarray analysis can detect chromosomal abnormalities in cells obtained from an invasive procedure [24]. Prenatal testing for fetal chromosomal abnormalities aims at ascertaining which women are at a higher risk of carrying an affected fetus, thus enabling the parents to reach informed decisions as to whether to proceed with further diagnostic testing. Clinically significant fetal chromosomal abnormalities involve losses or gains of genetic material ranging from small segments of chromosomes to entire chromosomes. However, despite the importance of undergoing invasive procedures during pregnancy, women from diverse ethnic groups espouse different attitudes and beliefs regarding these tests [24]. 

Our aim was to determine whether there are any disparities in women’s attitudes towards performing numerous prenatal tests and pregnancy terminations when genetic HL is detected compared with women afflicted with non-genetic HL. We also compared two ethnic groups—Jewish and Muslim women—and examined whether further support from medical professionals, especially genetic counselors, nurses, or medical doctors, would be needed in the future. 

## 2. Materials and Methods

### 2.1. Study Design and Setting

A cross-sectional study was performed among Israeli Jewish and Muslim women with various levels of religious observance, as well as age, socio-economic status, and academic achievement. A snowball and convenience sample were used. Following an advertisement posted on social media, the participants agreed to answer an online questionnaire (Appendix A). Inclusion criteria were age (at least 18 years old) and sex (females).

In order to assess the statistical differences across groups, chi-squared tests were employed. The results showed that there were statistically significant differences between the different nationality groups as regards CVS or AC: c^2^ (3, N = 953) = 76.28, *p* < 0.001, r_c_ = 0.28. This indicated that Muslim and Jewish individuals tended to respond “yes”, whereas Bedouins and Druze individuals tended to respond “no”. Additionally, the pregnancy termination/abort was as follows: c^2^ (3, N = 953) = 107.79, *p* < 0.001, r_c_ = 0.34. This indicated that Jewish individuals tended to respond “yes”, whereas Muslims, Bedouins, and Druze individuals tended to respond “no”.

### 2.2. Study Tool

An online questionnaire (Appendix A) was constructed and validated by five professional and experienced individuals in the field of research for adequate content validity. The questionnaire consisted of 25 items in total and was addressed specifically to women. Items included (1) demographical details (i.e., age, respondent’s residence, population group, religiosity level, and education level); (2) four yes/no questions—having first- or second-degree relatives afflicted with a hearing impairment or deafness/having a child afflicted with a hearing impairment or deafness/undergone invasive medical procedures to diagnose deafness or hearing impairment, i.e., CVS or AC, a pregnancy termination/abortion due to deafness or a hearing impairment diagnosis; (3) the rest of the items, which examined the extent of the participants’ perceived support and consultations regarding pregnancy terminations, all performed on a Likert scale between 1 (strongly disagree) and 5 (strongly agree).

### 2.3. Data Analysis

The assumption of the normal distribution was performed using the Kolmogorov–Smirnov test for the study variables. Continuous variables in our study were distributed normally and described as the mean and standard deviation (SD) using the Student’s *t*-test or one-way ANOVA tests; however, chi-squared or Fisher’s exact test were used for categorical variables. Pearson correlations were calculated to assess the association between the general opinion on pregnancy abortion and the general perceived support between the Muslim and Jewish groups. All statistical tests were two-sided, and *p* < 0.05 was considered statistically significant. Data analysis was performed using Statistical Package for the Social Science (SPSS) version 28 (IBM, Armonk, NY, USA).

### 2.4. Ethics Approval

The study protocol was approved by the Institutional Review Board (IRB) at the Academic College of Tel Aviv-Yaffo (Protocol number 2021-1087); all procedures were performed in accordance with local guidelines and regulations.

## 3. Results

Overall, there were 953 Israeli women respondents, aged 18–46 years (Mean = 32.00, SD = 7.12). Of those, 68.7% were city dwellers, and 31.3% were village dwellers. However, 42.9% were Muslims, 8.2% were Bedouins, 9.1% were Druze, and 39.8% were Jewish women. Nearly half (48.7%) had not received an academic education. All participants had a child with a hearing impairment or deafness. In addition, 42.8% have first- or second-degree relatives afflicted with a hearing impairment or deafness. Data regarding awareness of their partner or themselves being carriers of deafness-related genes/genetic diseases indicated that 38.8% were carriers of a connexin-related deafness gene, 17.2% carried nephron-phthisis genes, 7.5% carried Alport’s syndrome genes, 5.6% carried Usher’s syndrome genes, and 31% reported: “did not know”.

Significant differences in the carrying of deafness-related genes were found between the different population groups: c^2^ (4, N = 953) = 144.16, *p* < 0.001, r_c_ = 0.39. However, the Jewish participants carried more Usher’s syndrome genes (14%) than Muslim women (0%). Connexin-related deafness genes and nephronophthisis were more common in the Muslim (40.9%, 25.1%, respectively) than in Jewish participants (35.6%, 5.3%, respectively). However, no significant differences between population groups were found for Alport’s syndrome. 

In order to investigate probable hearing impairment in a fetus, 57.1% performed CVS or AC. However, 26.5% performed a pregnancy termination/abortion due to a hearing impairment or deafness of their fetus. Table 1 indicates the distribution of performing CVS or AC and pregnancy termination/abortion between different ethnic groups.

Table 2 indicated a significant association between the carrying of a hearing impairment gene and performing a CVS or AC test, both in the Muslim group: c^2^ (4, N = 574) = 187.08, *p* < 0.001, r_c_ = 0.57. This denoted that carriers of connexin-related deafness, nephronophthisis, and Alport’s syndrome did not undergo the abovementioned procedures and in the Jewish group: c^2^ (4, N = 379) = 144.52, *p* < 0.001, r_c_ = 0.62. This indicated that carriers of every gene (i.e., Usher’s syndrome, connexin-related deafness, nephronophthisis, and Alport’s syndrome) did not perform the medical procedures, whereas those who responded “did not know” tended to perform the procedures.

Carrying a hearing impairment gene was also significantly associated with a pregnancy termination/abortion. The results showed that there were statistically significant differences in the Muslim group: c^2^ (4, N = 574) = 248.06, *p* < 0.001, r_c_ = 0.66. These indicated that carriers of connexin-related deafness, nephronophthisis, and Alport’s syndrome did undergo a pregnancy termination/abortion (Table 2). City dwellers had a higher tendency to undergo a pregnancy termination/abortion than the village dwellers in the Muslim group: c^2^ (1, N = 574) = 44.24, *p* < 0.001, r_c_ = 0.28. However, since Jewish participants lived solely in cities, we were unable to compute or compare the results of this group, as there was not a second sub-group (i.e., only “city” without “village”). Secular individuals, those with an academic degree, and those who had first- or second-degree relatives with a hearing impairment or deafness tended to perform CVS/AC test or pregnancy termination/abortion more than religious and non-academic individuals or respondents without first- or second-degree relatives with a hearing impairment or deafness, both in the Muslim and the Jewish populations (Table 2).

A one-way ANOVA explored the differences across religiosity levels between the Muslim and Jewish participants as regards the three questions relating to pregnancy termination/abortion (genetic consultation, religious-figure consultation, and the linkage between religiosity and pregnancy termination). The results are displayed in Table 3, which show significant differences in all parameters between religious Muslim and Jewish populations vs. the secular populations. Post hoc analyses were determined and are also displayed in Table 3.

In order to explore the differences between the Muslim and the Jewish groups according to the responses to the three questions regarding pregnancy termination/abortion (genetic consultation, religious-figure consultation, and the linkage between religiosity and pregnancy abortion), the general opinion on abortion, the general perceived support and opinion as regards in vitro fertilization to avoid deafness, independent-samples *t*-tests were calculated. The results demonstrated the following: (1) Jewish individuals reported higher rates of genetic consultation than Muslim individuals; (2) no statistical differences were found between the groups (Jewish vs. Muslim) in seeking religious-figure consultations; (3) no statistical difference was found between the groups (Jewish vs. Muslim) as regards their opinion regarding the linkage between religiosity and pregnancy abortion; (4) Jewish individuals reported a higher general tendency to undergo a pregnancy termination and prenatal testing than the Muslim individuals; (5) Jewish individuals reported a higher general perception of support than the Muslim individuals; and (6) Jewish individuals reported higher tendencies to agree to in vitro fertilization and, thereafter, a PGD procedure to avoid deafness in the fetus than the Muslim individuals.

Finally, to assess the association between the general opinion on pregnancy abortion and the general perceived support between the Muslim and Jewish groups, zero-order Pearson correlations were calculated. The results indicated that the Muslim group generally perceived support positively and strongly correlated with the general opinion on pregnancy abortion, r(574) = 0.95, *p* < 0.001, indicating that an increase in support was followed by an increase in predisposition to undergo a pregnancy termination. Furthermore, the results indicated that the Jewish group, generally, positively perceived support and strongly correlated with the general opinion on pregnancy termination: r(379) = 0.98, *p* < 0.001. This indicated that an increase in support was followed by an increase in predisposition towards pregnancy termination.

## 4. Discussion

A significant increase in the number of genes causing autosomal recessive diseases among Israeli Muslims from 2010 to 2018 has been recorded in the Israeli National Genetic Database. The number of variants increased three-fold [25]. Consanguinity and marriages between Israeli Muslim relatives are a frequent phenomenon in the Middle East. At present, >150 genes have been recognized as causing HL, with specific genes contributing to deafness. Of these, >20 genes have been associated with deafness in the Jewish–Israeli hearing-impaired population [4]. In the Muslim population, at least 44 distinct genes have been implicated [8]. The most common gene found in families afflicted with deafness in our populations was the connexin-related deafness gene (also found in the literature) [4,8,26]. Families in both ethnic groups, Jews and Muslims, tended to perform more prenatal tests and terminate more abnormal pregnancies when HL was not genetically proven. It is well known that pregnancy termination is forbidden among Muslims after the fetus has reached a maximum development of 120 days. Most Muslims and Druze scholars agree that abortion is prohibited after the ensoulment at 120 days [10,11,12,13]. HL is not considered a serious disability in Islam and most Muslim women do not terminate their pregnancies [10]. In the Jewish population, most ultra-orthodox Jews do not terminate their pregnancies [27,28]. 

The religiosity in both ethnic groups strongly influences whether to undergo any invasive prenatal tests. In general, in both ethnic groups, secular women will receive more genetic counseling than religious women. Muslim religious women and most ultra-Orthodox Jews will not undergo these tests, whereas secular women in both groups and religious women in the Jewish group will. Similar results have been reported in the literature relating to both ethnic groups, although not specific to any of the HL illnesses. Furthermore, as found in the literature, religious individuals will less frequently terminate their pregnancies than secular women in both ethnic groups [27,28,29,30,31].

Our findings, in agreement with the literature, demonstrated that an academic education has a major influence on the woman’s decision to undergo or not undergo invasive prenatal tests in both study groups [32,33,34]. Individuals who received an academic education have more access to digital and general knowledge. In both ethnic groups, individuals with first- or second-degree relatives afflicted with HL or hearing impairment will undergo less invasive prenatal diagnosis procedures and pregnancy terminations than those without such relatives. The assumption is that those with relatives afflicted with HL can closely monitor this hearing loss, deemed to be very minor, which can be corrected in early childhood [35,36,37,38]. These individuals will not perform an invasive and risky procedure while pregnant and, hence, will not terminate their pregnancy. 

The differences between the two ethnic groups in this study were emphasized in the Jewish population who underwent more invasive prenatal procedures and demonstrated a general tendency toward terminating more abnormal pregnancies than the Muslim women, as previously reported [9,29]. Furthermore, much more support was given to the Jewish families, aiding in the decision-making as to whether to terminate the pregnancy [9]. It has been reported that in the Jewish secular population, the women aspire for a perfect child [39], which is probably the reason for the support given by close family and friends [40]. Secular Jewish women have more frequently undergone pregnancy terminations due to the fetus’s deafness. In Jewish secular society, it is socially acceptable to terminate a pregnancy [39]. 

Our results also showed that Jewish women underwent in vitro fertilization (IVF) more often and subsequently, a PGD, to avoid giving birth to a deaf baby than the Muslim women. Jewish women most likely receive more consultations as regards the IVF and PGD procedures and are more aware and willing to undergo these procedures. According to our results, they also receive more genetic counseling than Muslim women. More genetic counseling relating to IVF and PGD procedures should be given to the Muslim population. The PGD would prevent them from performing invasive and risky prenatal tests and does not require a pregnancy termination [41,42,43]. Muslim city-dwelling women tended to undergo more pregnancy terminations than village dwellers, where the religious influence was more extensive. We previously reported this finding as regards the performance of more prenatal tests and pregnancy terminations in women who lived in Muslim cities than in Muslim villages. In Muslim cities, there are more health services and counseling in every medical field. The Muslim women living in the cities are less religious compared to the village dwellers [44]. 

This is a pioneering and innovative study presenting different views on the undergoing of prenatal testing and pregnancy termination in Israeli and Muslim families afflicted with HL or non-genetic HL. We encourage further counseling and health assistance to the Muslim population as regards how to diagnose this impairment, make a diagnosis without pregnancy termination, and if possible, repair the impairment.

### Strength and Limitations

A strength of our study was that we were able to determine where and to whom to provide more counseling explanations and health professional involvement in both populations dealing with a deaf child.

The limitations of the current study focused on the further need to know the reasons why the families with a child afflicted with HL did not terminate their pregnancies in both ethnic populations. Moreover, we should have asked the participants if they were more afraid of giving birth to another child afflicted with various types of hearing loss and if they received medical information regarding IVF and PGD procedures.

## 5. Conclusions

No significant difference was found between the behavior of Jewish and Muslim women with a child afflicted with HL, genetic or unexplained, when undergoing prenatal invasive testing. Secular Jewish women have more frequently undergone pregnancy terminations due to the fetus’ deafness. Further genetic counseling and information concerning IVF and PGD procedures should be provided to the Muslim population. 

## Figures and Tables

**Table 1 children-10-01438-t001:** Distribution of performing chorionic villus sampling or amniocentesis and pregnancy termination/abortion by population group.

		Muslims		Bedouins		Druze		Christians	
		N	%	N	%	N	%	N	%
Performed chorionic villus sampling or amniocentesis	No	152	37.2	48	61.5	71	81.6	72	26.6
	Yes	257	62.8	30	38.5	16	18.4	199	73.4
Underwent a pregnancy termination/abortion	No	328	80.2	74	94.9	84	96.6	158	58.3
	Yes	81	19.8	4	5.1	3	3.4	113	41.7

**Table 2 children-10-01438-t002:** Factors associated with undergoing CVS/AC test or pregnancy termination/abortion by population group.

Categorical Variable 1	Categorical Variable 2	Population Group	c^2^ (df)	Corr.
Carrying a hearing impairment gene	Underwent CVS or AC testing	Muslim	187.08 (4) *	0.57
Carrying a hearing impairment gene	Underwent CVS or AC testing	Jewish	144.52 (4) *	0.62
Carrying a hearing impairment gene	Pregnancy termination/abortion	Muslim	248.06 (4) *	0.66
Settlement (cities vs. villages)	Pregnancy termination/abortion	Muslim	44.24 (1) *	0.28
Religiosity level	Underwent CVS or AC testing	Muslim	47.50 (1) *	0.29
Religiosity level	Underwent CVS or AC testing	Jewish	97.78 (2) *	0.51
Religiosity level	Pregnancy termination/abortion	Muslim	25.00 (1) *	0.21
Religiosity level	Pregnancy termination/abortion	Jewish	107.78 (2) *	0.53
Education level	Underwent CVS or AC testing	Muslim	17.31 (1) *	0.17
Education level	Underwent CVS or AC testing	Jewish	38.76 (1) *	0.32
Education level	Pregnancy termination/abortion	Muslim	9.17 (1) *	0.13
Education level	Pregnancy termination/abortion	Jewish	10.20 (1) *	0.16
Relatives with a hearing impairment	Underwent CVS or AC testing	Muslim	261.47 (1) *	0.68
Relatives with a hearing impairment	Underwent CVS or AC testing	Jewish	74.60 (1) *	0.44
Relatives with a hearing impairment	Pregnancy termination/abortion	Muslim	83.67 (1) *	0.38
Relatives with a hearing impairment	Pregnancy termination/abortion	Jewish	46.26 (1) *	0.35

* *p* < 0.001. Corr. = correlation coefficient. Population group: Muslims (N = 574), Jewish (N = 379). Religiosity: secular, traditional, religious, ultra-orthodox. Education level: non-academic, academic. Relatives with a hearing impairment: does not have a first- or second-degree relative with a hearing impairment, does have a relative with a hearing impairment.

**Table 3 children-10-01438-t003:** Differences between religiosity levels on three criteria for pregnancy termination/abortion among Muslim and Jewish groups.

Group	Criterion	Religiosity	N	M	SD	F-Test
Muslims	Genetic consultation	Secular	166	3.42 _a_	2.12	73.49 *, w^2^ = 0.11
		Religious	408	2.07 _b_	1.51	
	Religious-figure consultation	Secular	166	3.85 _b_	1.87	17.61 *, w^2^ = 0.03
		Religious	408	4.63 _a_	2.10	
	Link between religiosity and pregnancy termination	Secular	166	4.13 _b_	2.00	66.62 *, w^2^ = 0.10
		Religious	408	5.52 _a_	1.79	
Jewish	Genetic consultation	Secular	146	4.31 _a_	2.34	60.80 *, w^2^ = 0.24
		Religious	143	3.17 _b_	2.49	
		Orthodox	90	1.17 _c_	0.50	
	Religious-figure consultation	Secular	146	3.25 _b_	2.37	21.65 *, w^2^ = 0.10
		Religious	143	4.78 _a_	2.45	
		Orthodox	90	5.18 _a_	2.59	
	Link between religiosity and pregnancy abortion	Secular	146	3.84 _c_	2.51	56.38 *, w^2^ = 0.23
		Religious	143	4.85 _b_	2.49	
		Orthodox	90	6.94 _a_	0.23	

* *p* < 0.001. Lower-case letters reflect post hoc tests (Tukey’s HSD) as follows: (1) consecutive letters show lower means (e.g., a > b; b > c; a > b > c, etc.); (2) means that these share a letter, per comparison, and indicates no statistical difference between them. Notably, post hoc tests were not performed for the Muslim group as the number of categories is two (i.e., secular vs. religious), and, as such, required no additional post hoc testing.

## Data Availability

Individual-level data cannot be made publicly available due to legal and ethical restrictions. Aggregative data might be provided upon reasonable request to the corresponding author.

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
