# Peer review of "Prenatal Diagnosis and Pregnancy Termination in Jewish and Muslim Women with a Deaf Child in Israel"

_children, 2023, doi:10.3390/children10091438_

Round 1

Reviewer 1 Report

The Authors report an interesting study on the attitudes of various ethnic and religious groups living in Israel towards prenatal diagnosis and termination of pregnancy due to hearing loss.

Generally, the manuscript should be reread by a native English speaker familiar with medical terms.

Gene names by convention should be written in italics.

The title mentions families, but the study was conducted on women.

The division of the studied population into groups is not homogeneous throughout the manuscript (Jews, Jewish, Muslims, Muslim Arab, Bedouin Arab, Druze, Christian).

It is probably me who doesn't quite understand, the Authors mention different nationalities, but isn't the population studied all Israeli?

Hearing loss is called a malformation in many places in the manuscript, but this is absolutely incorrect. The term malformation has a precise meaning, and hearing loss is not a malformation.

The results of the study should be reported in a more orderly manner with the help of more tables.

Page 1, Abstract, line 1: The sentence "Deafness is the most sensory disability in humans" seems to be incomplete. The whole abstract is unclear and should be rewritten.

Page 2, second and third paragraphs: the text is confusing. While talking about prenatal diagnosis and prenatal screening for chromosomopathies, it suddenly switches to talking about the termination of pregnancy for deafness.

Page 7, Discussion, line 5: "HL loss" should be "HL".

Page 7, Discussion, line 11: "genetically approved" should be "genetically proven" or something similar.

The manuscript would benefit from a re-reading by a native English speaker familiar with medical terms.

Author Response

Point-by-point reply

We thank the editor and reviewer for the valuable comments and suggestions.

Reviewer 1

Comments and Suggestions for Authors

The Authors report an interesting study on the attitudes of various ethnic and religious groups living in Israel towards prenatal diagnosis and termination of pregnancy due to hearing loss.

Reply: We thank the reviewer for the positive comment.

Generally, the manuscript should be reread by a native English speaker familiar with medical terms.

Reply: Following the reviewer’s comment, we revised the text throughout the manuscript by a native English speaker. Please see the revised manuscript.

Gene names by convention should be written in italics.

Reply: Following the reviewer’s comment, we have rewritten the Gene names in italics throughout the manuscript. Please see the revised manuscript.

The title mentions families, but the study was conducted on women.

Reply: In view of the reviewer's comment, we changed the title accordingly to "Prenatal diagnosis and pregnancy termination in Jewish and Muslim women with a deaf child in Israel"

The division of the studied population into groups is not homogeneous throughout the manuscript (Jews, Jewish, Muslims, Muslim Arab, Bedouin Arab, Druze, Christian).

Reply: Following the reviewer’s comment, we united the terms in a homogeneous way by changing Jews to "Jewish", Muslims and Muslim Arab to "Muslim", Bedouin Arab to "Bedouins", throughout the manuscript. Please see the revised manuscript.

It is probably me who doesn't quite understand, the Authors mention different nationalities, but isn't the population studied all Israeli?

Reply: We apologize for the incorrect mention of the term "different nationalities". In view of the reviewer's comment, we modified it to "population group". Please see the revised manuscript.

Hearing loss is called a malformation in many places in the manuscript, but this is absolutely incorrect. The term malformation has a precise meaning, and hearing loss is not a malformation.

Reply: We apologize for the incorrect mention of Hearing loss as a malformation. In view of the reviewer's comment, we removed any mention of hearing loss as a malformation. Please see the revised manuscript.

The results of the study should be reported in a more orderly manner with the help of more tables.

Reply: We thank the reviewer for the important comment. Following the reviewer's suggestion, we have rewritten the methods and results sections in an orderly manner with tables. Please see the revised methods and results sections.

Page 1, Abstract, line 1: The sentence "Deafness is the most sensory disability in humans" seems to be incomplete. The whole abstract is unclear and should be rewritten.

Reply: Following the reviewer's suggestion, we have rewritten the abstract section and modified the sentence"Deafness is the most sensory disability in humans" to "Deafness is the most common sensory disability in humans, influencing all aspects of life, However, early diagnosis of hearing impairment and initiating the rehabilitation process is of great importance, in order to enable the development of language and communication as soon as possible". Please see the revised abstract section.

Page 2, second and third paragraphs: the text is confusing. While talking about prenatal diagnosis and prenatal screening for chromosome pathies, it suddenly switches to talking about the termination of pregnancy for deafness.

Reply: Following the reviewer's suggestion, we have rewritten the second and third paragraphs and modified them accordingly. Please see the revised introduction section, page 1, lines 38-41; and page 2, line 42, and lines 63-87.

Page 7, Discussion, line 5: "HL loss" should be "HL".

Reply: Following the reviewer's comment, we modified "HL loss" to "HL". Please see the revised discussion section, page 7, line 248.

Page 7, Discussion, line 11: "genetically approved" should be "genetically proven" or something similar.

Reply: Following the reviewer's comment, we modified "genetically approved" to "genetically proven" Please see the revised discussion section, page 7, line 254.

Reviewer 2 Report

Introduction:

More details regarding congenital hearing loss (with appropriate references) should be provided.

In particular, the importance of early diagnosis through neonatal hearing screening programmes should be explained.

Materials and Methods and Results:

these sections are often confused and not well-structured: the results should not be presented in the methods section and vice versa. Besides, the same concepts are repeated several times.

The online questionnaire that was sent to participants should be present in the work.

In the text several grammar errors are present, therefore accuracy should be improved.

Author Response

Reviewer 2

Introduction:

More details regarding congenital hearing loss (with appropriate references) should be provided.

Reply: Following the reviewer's suggestion, we have expanded the information regarding congenital hearing loss. Please see the revised introduction section, page 1, lines 38-41; and page 2, line 42. “GJB2 variants are the most prevalent cause of hereditary hearing loss worldwide and are responsible for approximately 30% of deafness in Jewish families [1-3]. In Israel, mutations include some responsible for hearing loss, such as GJB2 c.167delT and TMC1 p.Ser647Phe, while other deafness-causing mutations, such as GJB2 c.35delG, are common in all Jewish ethnicities and elsewhere [1, 4].

In particular, the importance of early diagnosis through neonatal hearing screening programmes should be explained.

Reply: Following the reviewer's suggestion, we have explained the importance of early diagnosis through neonatal hearing screening programs. Please see the revised introduction section, page 2, lines 63-87. “The importance of early identification of hearing impairment is well established. Cumulative evidence shows that undiagnosed or untreated permanent hearing impairment (PHI) during early childhood may result in speech-language delay, poor academic achievements, and social and emotional difficulties [5]. Such delays in the different domains have been documented also for those whose PHI was mild to moderate or only in one ear [6-8]. Currently, there is overwhelming evidence that early diagnosis and habilitation before the age of six months improves speech and language development and cognitive outcomes, reducing the need for special education and improving quality of life [9-11].  In order to identify hearing-impaired infants as early as possible and offer them the appropriate intervention, the National Institute of Health in 1993 recommended the implementation of universal neonatal hearing screening programs (NHSP) up to the age of three months in order to initiate hearing habilitation no later than the age of six months [12]. The Israeli national hearing screening program at that time was conducted at Mother and Child Health Clinics using the distraction test at ages 7–9 months. Since 1997, a number of medical centers in Israel began offering NHSP. The Ministry of Health Directive 33/2009 established the guidelines for the NHSP for all infants to be implemented from January 1st, 2010 [11-13]. Following the recommendations stated in these guidelines, the current program consists of the Otoacoustic Emissions (OAE) test for all newborns, and the Automated Auditory Brainstem Response test was established for those infants who failed OAE testing and for infants at risk for auditory neuropathy spectrum disorders [14]. The national program objectives are to complete screening by age one month, conduct diagnostic audiological testing no later than age three months for those infants who failed the screening, and initiate habilitation for those diagnosed with hearing loss by age six months. Early screening covers all newborns in the country and continues for those who failed the secondary screens and those identified as high-risk groups [14]”.    

Materials and Methods and Results:

these sections are often confused and not well-structured: the results should not be presented in the methods section and vice versa. Besides, the same concepts are repeated several times.

Reply: We thank the reviewer for the important comment. Following the reviewer's suggestion, we have rewritten the methods and results sections in an orderly manner with tables. Please see the revised methods and results sections.

The online questionnaire that was sent to participants should be present in the work.

Reply: Following the reviewer's suggestion, we have added an appendix with the online questionnaire as supplementary material, in addition regarding the questionnaire and the important items that were included in the methods section. Please see Appendix 1 as supplementary material.

Comments on the Quality of English Language

In the text several grammar errors are present, therefore accuracy should be improved.

Reply: We apologize for the several grammar errors. In view of the reviewer's comment, we revised the text throughout the manuscript by a native English speaker. Please see the revised manuscript.

References:

  1. Brownstein Z, Avraham KB. Deafness genes in Israel: implications for diagnostics in the clinic. Pediatr Res. 2009; 66(2): 128-134.
  2. Sobe T, Vreugde S, Shahin H, et al. The prevalence and expression of inherited connexin 26 mutations associated with nonsyndromic hearing loss in the Israeli population. Hum Genet. 2000; 106(1): 50-57.
  3. Estivill X, Fortina P, Surrey S, et al. Connexin-26 mutations in sporadic and inherited sensorineural deafness. Lancet. 1998; 351(9100): 394-398.
  4. Brownstein Z, Friedman LM, Shahin H, et al. Targeted genomic capture and massively parallel sequencing to identify genes for hereditary hearing loss in middle eastern families. Genome Biol. 2011; 12(9): R89.
  5. Harlor ADB, Bower C. Hearing assessment in infants and children: recommendations beyond neonatal screening. Pediatrics. 2009;124(4):1252–63.
  6. Kishon-Rabin L, Kuint M, Hildesheimer D, Ari-Even Roth D. Delay in auditory behaviour and preverbal vocalization in infants with unilateral hearing loss. Dev Med Child Neurol. 2015;57(12):1129–36.
  7. Lieu J. Permanent unilateral hearing loss (UHL) and childhood development. Curr Otorhinolaryngol Rep. 2018;6(1):74–81.
  8. Tharpe AM. Unilateral and mild bilateral hearing loss in children: past and current perspectives. Trends Amplif. 2008; 12:7–15.
  9. Ching TYC, Dillon H, Leigh G, Cupples L. Learning from the longitudinal outcomes of children with hearing impairment (LOCHI) study: summary of 5-year findings and implications. Int J Audiol. 2018;57(sup2): S105–11.
  10. Centers for Disease Control and Prevention. Hearing Loss in Children: Data and Statistics 2014https://www.cdc.gov/ncbddd/hearingloss/data.html. Accessed 15 Jun 2017.
  11. Israeli Ministry of Health. Guidelines for newborn screening for identification of congenital hearing loss, 2009 [Hebrew]. https://www.health.gov.il/hozer/mr05_2018.pdf http://www.health.gov.il/hozer/mr33_2009.pdf. Accessed 27 Dec 2016.
  12. World Population Review. Israel Population, 2018. http://worldpopulationreview.com/countries/israel-population/. Accessed 8 Mar 2018.
  13. Central Bureau of Statistics, State of Israel. Israel in Figures. https://www.cbs.gov.il/en/Pages/search/searchResultsIsraelnFigures.aspx. Accessed 14 Feb 2019.
  14. Wasser, J., Ari-Even Roth, D., Herzberg, O. et al. Assessing and monitoring the impact of the national newborn hearing screening program in Israel. Isr J Health Policy Res 8, 30 (2019). https://doi.org/10.1186/s13584-019-0296-6.
